# Functional Ambivalence of Dendritic Cells: Tolerogenicity and Immunogenicity

**DOI:** 10.3390/ijms22094430

**Published:** 2021-04-23

**Authors:** Ji-Hee Nam, Jun-Ho Lee, So-Yeon Choi, Nam-Chul Jung, Jie-Young Song, Han-Geuk Seo, Dae-Seog Lim

**Affiliations:** 1Department of Biotechnology, CHA University, 335 Pangyo-ro, Bundang-gu, Seongnam, Gyeonggi-do 13488, Korea; wl08gml03@naver.com (J.-H.N.); csy5900@naver.com (S.-Y.C.); 2Pharos Vaccine Inc., 14 Galmachiro 288 bun-gil, Jungwon-gu, Seongnam, Gyeonggi-do 13201, Korea; jhlee@pharosvaccine.com (J.-H.L.); ncjung@pharosvaccine.com (N.-C.J.); 3Department of Radiation Cancer Sciences, Korea Institute of Radiological and Medical Sciences, 75 Nowon-ro, Nowon-gu, Seoul 01812, Korea; immu@kcch.re.kr; 4Department of Food Science and Biotechnology of Animal Products, Sanghuh College of Life Sciences, Konkuk University, 120 Neungdong-ro, Gwangjin-gu, Seoul 05029, Korea; hgseo@konkuk.ac.kr

**Keywords:** dendritic cells, tolerogenicity, regulatory T cells, autoimmune disease, inflammatory disease, immunotherapy

## Abstract

Dendritic cells (DCs) are the most potent professional antigen-presenting cells (APCs) and inducers of T cell-mediated immunity. Although DCs play a central role in promoting adaptive immune responses against growing tumors, they also establish and maintain peripheral tolerance. DC activity depends on the method of induction and/or the presence of immunosuppressive agents. Tolerogenic dendritic cells (tDCs) induce immune tolerance by activating CD4^+^CD25^+^Foxp3^+^ regulatory T (Treg) cells and/or by producing cytokines that inhibit T cell activation. These findings suggest that tDCs may be an effective treatment for autoimmune diseases, inflammatory diseases, and infertility.

## 1. Introduction

Dendritic cells (DCs) are the most potent professional antigen-presenting cells (APCs) and are responsible for maintaining immune homeostasis [1,2]. In addition to inducing innate immune responses, DCs play a central role in inducing T cell-mediated immune responses. Immature DCs (imDCs) take up antigens and present them to naïve T cells, a process that induces DC maturation; by contrast, fully mature DCs (mDCs) promote adaptive immune responses by inducing effector T cells [3]. Active immunotherapy with mDCs affects adaptive immune responses to growing tumors, making mDCs an alternative to conventional cancer treatments [4]. In addition, the potency of DC-based immunotherapy can be increased by combining DCs with immune checkpoint inhibitors [5,6,7].

DCs are also responsible for establishing and maintaining peripheral immune tolerance. The intestinal immune system maintains a balance between responses to harmless bacteria and food antigens and immunity against pathogens [8,9,10]. Intestinal CD103^+^ DCs play a central role in regulating mucosal immunity via induction of CD4^+^CD25^+^Foxp3^+^ regulatory T (Treg) cells, which in turn inhibit cytokine production and reduce the functional activity of effector T cells [11,12].

Tolerogenic DCs (tDCs) show anti-inflammatory and immunosuppressive activity against various autoimmune diseases, including rheumatoid arthritis (RA) [13], experimental autoimmune myocarditis (EAM) [14], and acute myocardial infarction (AMI) [15]. Studies in animal models show that tDCs loaded with specific antigens ameliorate inflammation by activating Treg cells and/or by suppressing effector T cells [13,14,15]. Based on previous studies, various clinical trials have been conducted in patients with RA, type 1 diabetes, multiple sclerosis, and Crohn’s disease [16]. To date, however, few markers that distinguish tDCs from mDCs have been identified. There are many specific candidate markers of tDCs. Co-stimulatory molecules such as CD80 and CD86 are regarded as representative markers of tDCs. Indeed, expression of co-stimulatory molecules by tDCs and mDCs has been analyzed and compared; the results suggest that these molecules are not specific markers that can be evaluated independently. Moreover, cytokines produced by tDCs differ from those produced by mDCs. Anti-inflammatory cytokines, such as interleukin (IL)-4 and IL-10, are considered to be markers of tDCs, but these molecules are difficult to detect. Some biomolecules can be regarded as markers of tDCs. For example, high amounts of indoleamine 2,3-dioxygenase (IDO), an enzyme involved in tryptophan catabolism, are produced by CD103^+^ DCs in the mouse intestine [17]. Additionally, expression of complement subunit C1q may be a distinctive molecular marker of tDCs, although reports are contradictory. Thus, despite increasing knowledge about tDCs, a reliable marker remains elusive.

While attempting to identify potential markers of tDCs, we analyzed the gene expression profiles of different DC subsets [18]. Currently, we are attempting to correlate gene expression profiles with the function of these DC subsets. However, it is difficult to identify specific markers that regulate DC function. This review describes the distinct characteristics of tDCs and their roles in several diseases.

## 2. DCs Play a Central Role in Inducing Anti-Tumor Immune Responses

DCs play a key role in generating anti-tumor immune responses by presenting antigens to naïve T cells and inducing their differentiation into effector T cells (type 1 T helper (Th1) cells and cytotoxic T lymphocytes (CTL)). Presentation of antigen to T cells by DCs regulates anti-tumor immune responses through the immunological synapse by inducing different types of signals. DCs present antigens to naïve T cells expressing the major histocompatibility complex (MHC): T cell receptor complex (signal 1) and co-stimulatory molecules (signal 2). Activation of both signals induces activation and expansion of antigen-specific T cells [19,20,21]. DCs also produce additional signaling molecules (signal 3), which modify the different types of immune responses [22]. By signaling through the immunological synapse, DCs modulate differentiation of T cells that play a central role in adaptive immune responses. In addition, DCs are important activators of natural killer (NK) cells, which may play a critical role in eliminating virus-infected cells. DCs secrete IL-12, which activates NK cells; thus, DCs are critical for early anti-tumor immune responses.

Earlier studies report that mDCs produce high levels of pro-inflammatory cytokines, such as IL-12, IL-1β, IL-6, and tumor necrosis factor (TNF)-α, and that they express high levels of cell surface molecules such as MHC class II, CD40, and CD80/CD86. These cells play a crucial role in active immunotherapy by stimulating T cells. Despite the finding that previous use of immunotherapies, such as immune checkpoint inhibitors, and/or antibodies specific for anti-inflammatory molecules, provides objective survival benefits for patients, some of these treatments can have serious adverse effects related to excessive immune activation [23]. Combination immunotherapy with DC vaccines intensifies specific anti-tumor responses by increasing the CD4^+^ to CD8^+^ T cell ratio [5,6,7]. Although DC vaccines can induce effective anti-tumor responses, clinical results have not lived up to those observed in in vitro experiments and animal models. Dendritic cells also play a central role in embryo implantation by inducing maturation of uterine NK cells, tissue remodeling, and angiogenesis [24,25]. However, dysregulation of the balance between various DC subsets may disturb immune homeostasis, leading to infertility. During a normal pregnancy, maternal immunity maintains immune tolerance against the semi-allogeneic fetus. Expression of CD80/CD86, HLA-DR, and IL-12p70 by circulating DCs in pregnant women are lower than those in non-pregnant women [26]. During the early stages of pregnancy, Treg cells in draining lymph nodes migrate to the uterus and increase in number, resulting in induction of immunological tolerance. Thus, tDCs may reduce implantation failure rates by maintaining the balance among Th1, Th2, Th17, and Treg cells [27].

Although mDCs can induce antigen-specific immunogenic responses, excessive activation of the immune response leads to disorders of immune homeostasis. Other DC subtypes may be tolerogenic and therefore be responsible for establishing and maintaining immune tolerance. The activities of different DC subtypes may depend on their method of generation and/or on the presence of immunosuppressive factors.

## 3. Tolerogenicity of DCs

Immune cells, such as polymorphonuclear (PMN) leukocytes, monocytes, macrophages, and DCs, play key roles in protecting the host from aggressive inflammation triggered by pathogens or self-antigens [28]. Neutrophils, also known as PMN leukocytes, are the most abundant leukocyte in humans and are present in large numbers at sites of autoimmune damage. During inflammatory responses, neutrophils not only act as effector cells through phagocytosis and the production of lytic enzymes, reactive oxygen species and inflammatory mediators; but interact with other immune cells, such as NK cells, macrophages, DCs, T cells and B cells [29,30]. However, under low density conditions, neutrophils show immunosuppressive activity [31]. Likewise, new evidence has emerged indicating that circulating monocytes and tissue-resident macrophages have phenotypic and functional heterogeneity. Although CD14^+^CD16^+^ monocytes and Ly6C^high^ macrophages promote tissue damage and aggravate disease symptoms, in experimental models of inflammation and autoimmune disease, an increased frequency of CD14^low^CD16^+^ monocyte and Ly6C^low^ macrophages has been observed in models responding to therapy [32,33]. In addition, some tissue-resident conventional DCs or specialized subsets of DCs, such as Langerhans cells (CD207^+^) in the skin or CD103^+^ DCs in the mucosa, have critical roles in maintaining immune tolerance by increasing the Treg cell population and promoting the production of anti-inflammatory cytokines [34].

CD103^+^ DCs, which play a decisive role in maintaining mucosal immune homeostasis, possess immunosuppressive activity. Epithelial surfaces are continuously exposed to various antigens, including dietary materials, commensal bacteria, pathogenic viruses, and allergens. Goblet cells, which are specialized epithelial cells located in the small intestine, present antigens to CD103^+^ DCs, which then transport antigens to naïve T cells in Peyer’s patches [11]. Tolerogenicity of CD103^+^ DCs is essential for preventing immune responses against harmless materials, such as foods and/or commensal bacteria; otherwise, there is a real risk of a continuous and excessive immune response. Although CD103 is regarded as a marker of tDCs, accurate expression patterns and functions remain unclear. Furthermore, surface molecules CD80 and CD86 are regarded as markers of DC maturation. These co-stimulatory molecules bind to CD28 and cytotoxic T lymphocyte-associated protein-4 (CTLA-4) (CD152) on T lymphocytes, thereby regulating T cell activation [35]. Although both CD80 and C86 bind to CD28 and CTLA-4, the latter molecules trigger opposing immune responses. Binding of CD80 and CD86 to CD28 increases T cell activation by promoting differentiation of T cells into IFN-γ-producing type 1 T helper (Th1) cells and IL-4-producing type 2 helper (Th2) cells [36]. By contrast, binding of CD80 and CD86 to CTLA-4, which is expressed on Tregs, negatively regulates immune responses by removing CD80 and CD86 from DCs, a process that inhibits CD28-mediated stimulation of other effector T cells [37]. Moreover, DCs exposed to immunosuppressive agents that block DC maturation show lower expression of CD80 and CD86 than mDCs [13]. These findings suggest that CD80 and CD86 are potential markers that are specific for different DC subsets; however, they are not indicative of DC maturation status.

Studies in animal models of RA, EAM, and AMI, show that tDCs exert therapeutic effects [13,14,15]. Based on previous results, clinical trials have been conducted in patients with RA, type 1 diabetes, multiple sclerosis and Crohn’s disease [38,39,40]. Although tDC-based immunotherapy has potential, the mechanisms underlying its immunomodulatory activity and the specific markers expressed by DCs remain unclear. Further studies of gene and protein expression are needed to identify markers specific for tDCs and to fully understand their functions in immune suppression.

## 4. DCs Are Responsible for Peripheral Tolerance, But This Depends on the Method of Generation and/or the Presence of Immunosuppressive Agents

To generate imDCs reproducibly, a previous study cultured mouse bone marrow-derived monocytes for 8 days in the presence of granulocyte-macrophage colony-stimulating factor (GM-CSF) and IL-4. These imDCs, in turn, were cultured for 24 to 48 h with specific antigens and inflammatory cytokines to generate mDCs. Interestingly, shortening the incubation time to about 4 h, generated cells expressing lower levels of CD80 and CD86 than fully mature DCs; these “semi-mature” DCs had tolerogenic activity [41]. Similarly, treatment of DCs with immunosuppressive agents, such as rosiglitazone, resulted in downregulation of CD80 and CD86 and generated immunotolerant DCs. Rosiglitazone, a ligand of peroxisome proliferator-activated receptor-γ (PPAR-γ), suppressed activation of extracellular-signal-regulated kinase (Erk) 1/2, pp38, and nuclear factor kappa light chain enhancer of activated B cells (NF-κB) in DCs, which confirms the tolerogenic activity of these cells [13]. In addition to culture conditions, regulating signaling pathways responsible for production of immunosuppressive factors may also lead to immune tolerance.

Human tDCs can be induced from peripheral blood monocytes by treatment with GM-CSF and IL-4, together with immunosuppressive factors and/or anti-inflammatory cytokines. Among the various immunosuppressive factors, vitamin D, dexamethasone, and rapamycin have been examined in animal disease models and humans. Anti-inflammatory cytokines, such as IL-10 and transforming growth factor-β (TGF-β), have also been used to induce tDCs [16]. In the presence of these factors and cytokines, DCs express lower levels of co-stimulatory molecules and MHC class II, and produce higher levels of IL-10 (which inhibits expression of Th1 cytokines), than mDCs [42,43].

## 5. tDCs Induce Naïve T Cells Differentiation into Treg Cells

Regulatory T (Treg) cells play key roles in maintaining peripheral tolerance, which regulates the development of various autoimmune diseases. Treg cells can be classified into two groups based on their developmental origin; thymus (tTregs) and periphery (pTregs). tTregs, which express Foxp3 and have T-cell receptors (TCRs) of relatively high affinity, are predominant in the bloodstream and lymph nodes. These cells are mainly involved in providing tolerance to autoantigens [44]. pTregs, which express Foxp3 induced by IL-2 and TGF- β, are the most common in the peripheral tissue and mainly involved in peripheral inflammation against exogenous pathogens. Stable expression of Foxp3 plays a key role in maintaining the function of Tregs [45].

Herein, we will mainly focus on CD4^+^Foxp3^+^ Treg cells. CD4^+^CD25^+^Foxp3^+^ Treg cells maintain immunological tolerance to self-antigens by secreting immunosuppressive cytokines, thereby suppressing the development and proliferation of effector immune cells, such as activated CD4^+^-T and CD8^+^-T cells. Immunosuppressive cytokines, such as TGF-β, IL-10, and IL-35, inhibit synthesis of pro-inflammatory cytokines and, therefore, expansion of effector T cells [46]. Treatment of Th1 cells with TGF-β1 increases production of IL-10, which inhibits cytokine production and reduces the activity of effector T cells. In addition, Treg cells attenuate immune responses by modulating activation of immune cells. The negative regulator CTLA-4 is the best characterized membrane-associated molecule expressed by Treg cells. Treg-derived CTLA-4 induces production of IDO and suppresses activation of APCs by binding to CD80 and CD86 [47]. Although Treg cells are essential for maintaining immune homeostasis, these cells also promote immune escape from anti-tumor therapies, and accelerate tumor progression [48]. Tregs in the tumor microenvironment show high expression of programmed death receptor-1 (PD-1), an immune checkpoint that promotes apoptosis of activated T cells [49]. Thus, the anti-tumor immune responses resulting from depletion or functional suppression of Tregs are enhanced when combined with immune checkpoint inhibitors, or anti-CD25 agents [50,51]. Because Treg cells suppress anti-tumor responses, appropriate regulation is important for successful anti-tumor therapy. By contrast, Treg cells are effective as a treatment for autoimmune and inflammatory diseases [52,53,54]. Our previous studies confirmed that antigen-specific Treg cells effectively ameliorate inflammation in mouse models of EAM, collagen-induced arthritis (CIA), and myocardial infarction (MI) [13,14,15]. In addition, clinical trials show that injection of autologous tDCs, along with pharmacologic drugs, not only inhibits activation of effector T cells and increases Treg cell populations, but also regulates expression of co-stimulatory molecules. These Tregs are derived from naïve T cells through the action of tDCs, indicating that tDCs are involved in regulating immune responses. In autoimmune diseases, tDCs boost the properties of Tregs by shifting them from a pathogenic to a suppressive phenotype [55,56]. Although Tregs control systemic immune responses, we found that pulsing DCs with specific antigen increases the number of antigen-specific Tregs, suggesting that regulating the systemic immune responses of patients may increase the risk of pathogen infection [13,14,15]. Conversely, antigen-specific Treg cells may be important for treating autoimmune diseases.

## 6. Identifying Reliable Markers of tDCs Is Important

Suboptimal maturation and tolerogenicity of DCs may contribute to limited clinical efficacy. Despite tDCs having immunosuppressive properties, markers specific for immunosuppressive tDCs have yet to be identified; although surface molecules, anti-inflammatory cytokines, IDO, and some other molecules have been suggested. Overall, tDCs express relatively lower levels of MHC class II and co-stimulatory molecules than mDCs. Additionally, tDCs show high expression of anti-inflammatory molecules (e.g., IL-4, IL-10, and TGF-β), and lower expression of pro-inflammatory molecules, than mDCs. Moreover, immunosuppressive cells and the tumor microenvironment show high expression of IDO. IDO induces immunosuppressive activity in cells by reducing their tryptophan concentration and producing immunomodulatory tryptophan metabolites [57]. This process inhibits T cell proliferation and activates mature Treg cells [58]. Recently, the complement subunit C1q was identified as a marker of tDCs. C1q suppresses activation of CD4+ T cells by increasing IL-10 secretion [59], and it inhibits differentiation and activation of DCs [60]. Although there are many candidate molecules that may be specific markers of tDCs, no dependable marker has emerged. Microarray analysis of functionally classified tDCs and mDCs, shows differences in the gene expression profiles of these DC subsets [18]. Currently, we are investigating the gene expression profiles and functions of different DC subsets, and assessing whether the therapeutic effects observed in mouse disease models correlate with those in human autoimmune diseases. Other important topics include the effect of selected markers on disease activity, as well as the effects of tDCs on the immune system in general.

## 7. tDCs Have Therapeutic Effects in Autoimmune and Inflammatory Diseases

While the normal immune system specifically recognizes and removes foreign pathogens to protect the host against infection, sometimes it causes autoimmune disease by mistakenly recognizing itself as an invader and attacking the host’s healthy cells and tissues. Historically, autoimmune disease has been regarded as an abnormal response. However, many studies have reported that autoimmune disease is a normal phenomenon, with self-reactive antibodies and cells present in all normal individuals [61]. During T cell development, T cell precursors differentiate into CD4^+^ and CD8^+^ T cells in the cortex of the thymus. These cells undergo a selection process in the thymus to ensure that TCRs lose their ability to recognize MHC molecules presenting its own antigens, or rather it has moderate affinity; this process is referred to as positive selection or MHC restriction [62,63,64]. Selected CD4 and CD8 T cells migrate into the medulla, and then autoreactive T cells are removed by interacting with APCs; this process is referred to as negative selection [65]. If autoreactive immune cells are not properly eliminated through these processes, autoimmunity may occur. Self-reactive immune responses are often induced in the process of mounting an immune response to foreign antigens [61]. About 3–5% of the world’s population is influenced by one of more than 80 different autoimmune diseases, with type 1 diabetes, multiple sclerosis, RA, systemic lupus erythematosus and Crohn’s disease being the most common of these conditions. The incidence and prevalence of autoimmune disease differ according to age, gender, ethnicity and geographical region. Over the past decade, the development of evidence-based technologies in molecular immunology and clinical laboratory testing have led to significant advances in diagnosis, disease classification, and prognosis [66]. The development of autoimmune disease depends on the properties of genes and environmental factors. Some genes affect the overall response of the immune system, so individuals are more susceptible to various types of autoimmune diseases. The signals from the environment affect the overall response of the immune system, the antigen-specific response, and the state of potential target tissues, influencing the development of autoimmunity. Various factors have been suspected to trigger autoimmunity, such as hormones, nutrition, pharmaceutical drugs, and toxins [61,67]. Most autoimmune diseases are more prevalent in women than in men. Sex hormones, such as estrogens and androgens, directly regulate the proliferation and activation of immune cells via surface receptors or intracellular receptors. Men often experience an aggressive inflammatory reaction, whereas women usually show greater antibody responses. The gender bias in autoimmune disease has attracted considerable attention, but the basis for this bias is not currently known [68].

Antibody therapy attenuates symptoms of autoimmune diseases, temporarily blocking specific molecules expressed by effector cells, and/or by regulating T cell developmental pathways [69,70]. Many therapeutic antibodies have provided a breakthrough for treatment of autoimmune disease [71]. Anti-CD25, anti-IL-2Rα, anti-CD52, anti-CD20, anti-IL-6R, and anti-TNF antibodies have been approved for clinical use against autoimmune diseases, such as multiple sclerosis, RA, and Chron’s disease [72]. However, although antibodies show therapeutic efficiency, they do not alter gene or protein expression. The therapeutic efficacy of tDCs was demonstrated in clinical trials of autoimmune diseases, such as RA, type 1 diabetes, multiple sclerosis, and Crohn’s disease. Administration of tDCs induces functional tolerogenic efficacy, as demonstrated by a decrease in effector T cell numbers and pro-inflammatory cytokine production, and an increase in the Treg cell populations [16]. Higher numbers of Tregs suppress the activation of CD4^+^- and CD8^+^-effector T cells by suppressing TCR-mediated NFAT and NF-κB signaling, producing anti-inflammatory cytokines, and reducing production of IL-2. In addition, maintaining immune tolerance in the uterus is necessary for a successful pregnancy; indeed, disruption of maternal tolerance is implicated in infertility [26,27]. Thus, establishing proper immune homeostasis could be a new goal for infertility treatments. At present, however, we lack detailed knowledge about the association between DCs and infertility. Despite a number of successes, the major challenge facing tDC-based immunotherapy is to optimize the protocol to obtain functionally stable tDCs. Currently on-going clinical trials of tDCs are focusing on standardization of methods used to generate tDCs ex vivo. Moreover, optimization of the dose, injection route, and frequency of administration, is necessary for successful application of tDCs.

## 8. Conclusions

DCs play a key role in inducing T cell-mediated immune responses; indeed, the maturation status of DCs is a critical determinant of their functional capacity. mDCs play a central role in promoting anti-tumor immune responses by increasing tumor-specific effector T cell populations. DCs are also important activators of NK cells, which may play a critical role in eliminating virus-infected cells. Although the number of immunogenic cells determines the tumor-specific immune response; the immunosuppressive cells and molecules produced by tumor cells, establish immune tolerance in the tumor microenvironment. The combination of DC-based immunotherapy with other tumor therapies, such as immune checkpoint inhibitors and/or immune suppressive molecule-specific antibodies, may overcome the limitations of cell therapy and increase their safety.

By contrast, tDCs are essential attenuators of destructive immune responses against harmless commensal bacteria and food components. Additionally, tDCs drive differentiation of native T cells to Treg cells, thereby preventing excessive activation of immune responses. Thus, the tolerogenicity of tDCs plays a central role in establishing and maintaining immune homeostasis. However, tDCs may also be effective in the treatment for autoimmune diseases, including rare diseases for which no treatment is currently available. Successful treatment with tDCs may require presentation of a specific antigen, leading to generation of antigen-specific Tregs that can suppress autoimmune responses. In addition, optimization of protocol design is required to obtain a number of stable tDCs.

Taken together, the findings presented in this review suggest that tDCs are essential regulators of immune responses. tDC-based immunotherapy for autoimmune and inflammatory diseases has potential as a novel therapeutic strategy for the future (Figure 1).

## Figures and Tables

**Figure 1 ijms-22-04430-f001:**
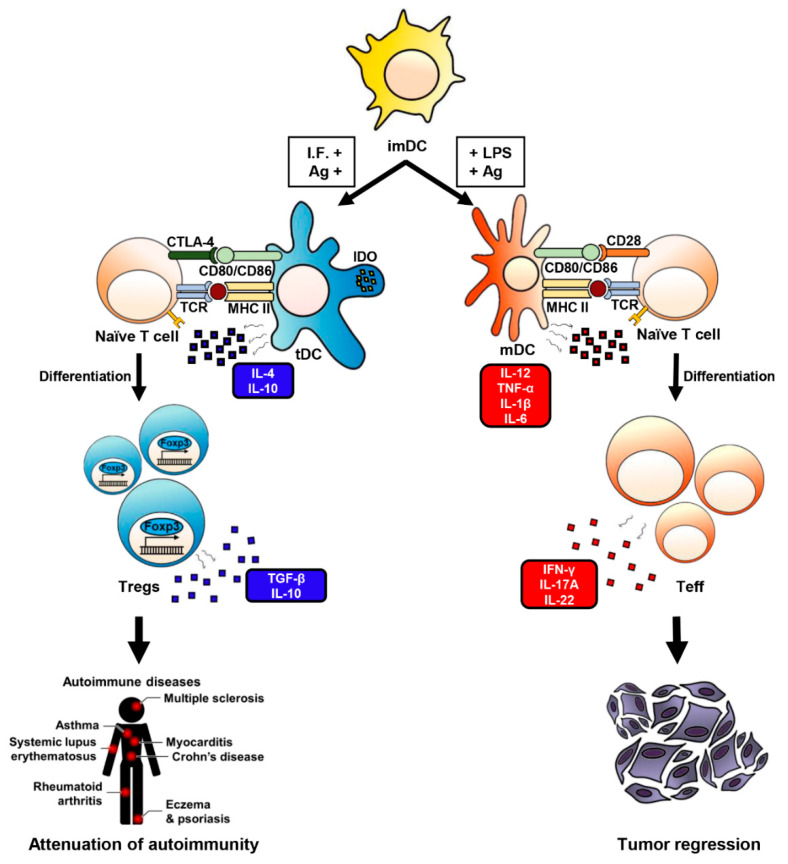
Functional overview of DCs. Generation and therapeutic effects of Treg cells and T helper cells. Abbreviations: imDC (immature DC); I.F. (immunosuppressive factor); Ag (antigen); LPS (lipopolysaccharide); NF-κB (nuclear factor kappa-light-chain-enhancer of activated B cells); PPAR-γ (peroxisome proliferator-activated receptor gamma); tDC (tolerogenic DC); mDC (fully-matured DC); IDO (indoleamine 2,3-dioxygenase); CTLA-4 (cytotoxic T-lymphocyte-associated protein 4); MHC II (major histocompatibility complex class II); TCR (T-cell receptor); Treg (T regulatory cell); Teff (effector T cell); Foxp3 (forkhead box P3); TGF-β (tumor growth factor-β).

## Data Availability

Not applicable.

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
