# Peer review of "Functional Ambivalence of Dendritic Cells: Tolerogenicity and Immunogenicity"

_ijms, 2021, doi:10.3390/ijms22094430_

Round 1

Reviewer 1 Report

The review entitled “Functional ambivalence of dendritic cells: Tolerogenicity and 2 immunogenicity” by Nam et al., is well constructed.

The authors have provided the basic concepts of tolerogenic DCs (tDCs) and immunogenic DCs with factors affecting their origin and functions. The authors also well described the role of tDCs in autoimmune inflammatory diseases and tumour and infectious diseases. The concept of tDCs induced Treg mediated immune responses is apparent.

The figure is nicely depicted.

This broad reference should be cited in the immunosuppressive therapies against autoimmune diseases. https://doi.org/10.1016/j.jaut.2018.08.009

Hereby I endorse the manuscript for publication.

Author Response

[15th, Apr. 2021]

Title: Functional ambivalence of dendritic cells: Tolerogenicity and immunogenicity

(Manuscript ID: ijms-1177677)

Dr. Gracie Zhang,

Editor-in-Chief,

Dear Dr. Zhang:

We are grateful to receive the comments of the reviewers regarding our manuscript entitled, “Functional ambivalence of dendritic cells: Tolerogenicity and immunogenicity”. This manuscript expounds the basic concepts of tolerogenic dendritic cells and immunogenic dendritic cells in autoimmune diseases and inflammatory diseases.

We appreciate the feedback provided by the reviewers and believe that this has helped us to improve our manuscript. We modified our manuscript according to all suggestions of the reviewers, and our point-by-point responses are listed below. Changes in the manuscript are highlighted by using “Track changes”.

We hope that the revised version will now be deemed suitable for publication in the International Journal of Molecular Sciences.

Thank you for your consideration. I look forward to receiving a response at your earliest convenience. 

Sincerely,

Dae-Seog Lim, PhD

Department of Biotechnology, CHA University, 335 Pangyo-ro, Bundang-gu, Seongnam, Gyeonggi-do 13488, Republic of Korea. Phone: +82-31-881-7227; Fax: +82-31-881-7228; E-mail: dslim@cha.ac.kr 

Responses to Reviewer 1

Thank you for your reviewing our manuscript and providing valuable comments. Our responses to each of these comments are listed below. We hope that our responses and actions will satisfy all your concerns, and that you will now consider the manuscript suitable for publication in the International Journal of Molecular Sciences.

Comments

The review entitled “Functional ambivalence of dendritic cells: Tolerogenicity and immunogenicity” by Nam et al., is well constructed.

The authors have provided the basic concepts of tolerogenic DCs (tDCs) and immunogenic DCs with factors affecting their origin and functions. The authors also well described the role of tDCs in autoimmune inflammatory diseases and tumour and infectious diseases. The concept of tDCs induced Treg mediated immune responses is apparent.

The figure is nicely depicted.

  1. This broad reference should be cited in the immunosuppressive therapies against autoimmune diseases. https://doi.org/10.1016/j.jaut.2018.08.009

Reply

Thank you very much for your comments. As you recommended, we added reference as well as the text as follows. The added reference number is 73. Please see below the “Modified text and Added reference”. Thank you again for your valuable comment.

Modified Text: In the page 6, Line 288; Section No. 7

Anti-CD25, anti-IL-2Rα, anti-CD52, anti-CD20, anti-IL-6R, and anti-TNF antibodies have been approved for clinical use against autoimmune diseases, such as multiple sclerosis, RA, and Chron’s disease [73].

Added reference: Reference No. 73

  1. R. Bonam, F. Wang, S. Muller. (2018). Autophagy: A new concept in autoimmunity regulation and novel therapeutic option. J Autoimmun. 94, 16-32.

Reviewer 2 Report

The manuscript by Nam et al. is a concise, but helpful summary of DC functions with respect to immunogenicity and tolerance. The only concern is that this relatively short, 5.5-page review has seven (!) authors and there are serious questions about contribution of four of them (hence this Reviewer has checked the box regarding ethical concerns). Much longer and exhaustive reviews in major journals usually have 3-4 authors. Otherwise, the review is coherent and easy-to-read. There are a few minor deficiencies in the text, which need to be corrected prior to publication and the authors are encouraged to reread their manuscript once again before re-submission in order to bring the number of these defects to a minimum. A non-exhaustive list of necessary corrections is as follows.

  1. Replace 'foodstuffs' with 'food antigens' (line 40).
  2. Replace 'previous studies' with 'earlier studies' (lines 84, 159).
  3. The sentence 'Under certain conditions, low-density granulocytes show immunosuppressive activity, referred to as PMN' (lines 117-118) makes no sense. Is activity 'referred to as PMN'? What is PMN, which come out of nowhere (and are not listed among the abbreviations in the end of the manuscript. Please correct.
  4. Change the sentence on line 180 to 'induce naïve T cells differentiation into Treg cells.'
  5. Change 'peripherally' to 'periphery' (line 183).
  6. 'The most' (line 187).
  7. Change the sentence on lines 214-215: 'These Tregs are derived from naïve T cells through the action of tDCs, suggesting that tDCs are involved in regulating immune responses.' tDC definitely regulate immune responses, this is not a 'suggestion', but a plain fact.
  8. Correct to 'Tregs' (plural) on line 219.
  9. The opposition 'abnormal' - ' natural' on lines 248-249 is incorrect. Please select the appropriate terms.
  10. Change 'current' to 'currently' (line 298).
  11. The last sentence 'Although tDC-based clinical trials are much smaller than those of immunogenic DCs, tDC-based treatments for autoimmune diseases may be a novel therapeutic strategy for the future' (lines 301-303) is illogical. Size of the trial does not correspond to novelty in any way. This sentence is also repeated in the end of Conclusions section (lines 326-328). Please correct and eliminate redundancy.
  12. What kind of contribution is 'collected the presented idea' (line 336)? Maybe, analyzed data? There is no data collection here since this is a review.  It's also unclear (see the major criticism at the start of this review) what kind of 'support from S.-Y.C., N.-C.J., J.-Y.S., and H.G.S.' was received over the writing of this relatively short manuscript. If the support was editorial, these individuals should be excluded from authors list and should be thanked in the Acknowledgements.

Author Response

[15th, Apr. 2021]

Title: Functional ambivalence of dendritic cells: Tolerogenicity and immunogenicity

(Manuscript ID: ijms-1177677)

Dr. Gracie Zhang,

Editor-in-Chief,

Dear Dr. Zhang:

We are grateful to receive the comments of the reviewers regarding our manuscript entitled, “Functional ambivalence of dendritic cells: Tolerogenicity and immunogenicity”. This manuscript expounds the basic concepts of tolerogenic dendritic cells and immunogenic dendritic cells in autoimmune diseases and inflammatory diseases.

We appreciate the feedback provided by the reviewers and believe that this has helped us to improve our manuscript. We modified our manuscript according to all suggestions of the reviewers, and our point-by-point responses are listed below. Changes in the manuscript are highlighted by using “Track changes”.

We hope that the revised version will now be deemed suitable for publication in the International Journal of Molecular Sciences.

Thank you for your consideration. I look forward to receiving a response at your earliest convenience. 

Sincerely,

Dae-Seog Lim, PhD

Department of Biotechnology, CHA University, 335 Pangyo-ro, Bundang-gu, Seongnam, Gyeonggi-do 13488, Republic of Korea. Phone: +82-31-881-7227; Fax: +82-31-881-7228; E-mail: dslim@cha.ac.kr 

Responses to Reviewer 2

Thank you for your reviewing our manuscript and providing valuable comments. Our responses to each of these comments are listed below. We hope that our responses and actions will satisfy all your concerns, and that you will now consider the manuscript suitable for publication in the International Journal of Molecular Sciences.

Comments

The manuscript by Nam et al. is a concise, but helpful summary of DC functions with respect to immunogenicity and tolerance. The only concern is that this relatively short, 5.5-page review has seven (!) authors and there are serious questions about contribution of four of them (hence this Reviewer has checked the box regarding ethical concerns). Much longer and exhaustive reviews in major journals usually have 3-4 authors. Otherwise, the review is coherent and easy-to-read. There are a few minor deficiencies in the text, which need to be corrected prior to publication and the authors are encouraged to reread their manuscript once again before re-submission in order to bring the number of these defects to a minimum. A non-exhaustive list of necessary corrections is as follows.

  1. Replace 'foodstuffs' with 'food antigens' (line 40).

Reply

Thank you very much for your comments. We changed the text as suggested. Thank you again for your valuable comment.

Modified Text: In the page 1, Line 40; Introduction section

The intestinal immune system maintains a balance between responses to harmless bacteria and food antigens and immunity against pathogens [8-10].

  1. Replace 'previous studies' with 'earlier studies' (lines 84, 159).

Reply

Thank you very much for your helpful comments. As you recommended, we changed the text. Thank you very much for your careful comment.

Modified Text: In the page 2, Line 84; Section No. 2

Earlier studies report that mDCs produce high levels of pro-inflammatory cytokines, such as IL-12, IL-1β, IL-6, and tumor necrosis factor (TNF)-α, and that they express high levels of cell surface molecules such as MHC class II, CD40, and CD80/CD86.

  1. The sentence 'Under certain conditions, low-density granulocytes show immunosuppressive activity, referred to as PMN' (lines 117-118) makes no sense. Is activity 'referred to as PMN'? What is PMN, which come out of nowhere (and are not listed among the abbreviations in the end of the manuscript. Please correct.

Reply

Thank you very much for your comments. Polymorphonuclear (PMN) leukocytes are a type of white blood cells include neutrophils, eosinophils, and basophils. The term polymorphonuclear leukocyte often refers specifically to “neutrophil granulocytes” the most abundant of the granulocytes. As you recommend, we changed some sentences to the appropriate term. Also, we added abbreviations in the end of manuscripts. Pleas below the “Modified text and added abbreviation” and reconsider it again. Thank you for your valuable comment.

Modified Text: In the page 3, Line 111-121; Section No. 3

Immune cells, such as polymorphonuclear (PMN) leukocytes, monocytes, macrophages, and DCs, play key roles in protecting the host from aggressive inflammation triggered by pathogens or self-antigens [28]. Neutrophils, also known as PMN leukocytes, are the most abundant leukocyte in humans and are present in large numbers at sites of autoimmune damage. During inflammatory responses, neutrophils not only act as effector cells through phagocytosis and the production of lytic enzymes, reactive oxygen species and inflammatory mediators, but interact with other immune cells such as NK cell, macrophages, DCs, T cells and B cells [29,30]. However, under low density condition, neutrophils show immunosuppressive activity [31]. Likewise, new evidence has emerged indicating that circulating monocytes and tissue-resident macrophages have phenotypic and functional heterogeneity.

Added Abbreviation: In the page 10, Line 352; Abbreviations section

PMN; Polymorphonuclear

  1. Change the sentence on line 180 to 'induce naïve T cells differentiation into Treg cells.'

Reply

Thank you very much for your helpful comments. We changed the text as suggested. Thank you very much for your helpful comment.

Modified Text: In the page 4, Line 183; Title of 5th section

tDCs induce naïve T cells differentiation into Treg cells

  1. Change 'peripherally' to 'periphery' (line 183).

Reply

Thank you very much for your helpful comments. As you recommended, we changed the text. Thank you very much for your valuable comment.

Modified Text: In the page 4, Line 187-188; Section No. 5

Treg cells can be classified into two groups based on their developmental origin; thymus (tTregs) and periphery (pTregs).

  1. 'The most' (line 187).

Reply

Thank you very much for your helpful comments. We changed the text as suggested. Thank you very much for your careful comment.

Modified Text: In the page 4, Line 191; Section No. 5

pTregs, which express Foxp3 induced by IL-2 and TGF- β, are the most common in the peripheral tissue and mainly involved in peripheral inflammation against exogenous pathogens.

  1. Change the sentence on lines 214-215: 'These Tregs are derived from naïve T cells through the action of tDCs, suggesting that tDCs are involved in regulating immune responses.' tDC definitely regulate immune responses, this is not a 'suggestion', but a plain fact.

Reply

Thank you very much for your comments. As you recommended, we modified the sentence. Please see below the “Modified Text” and consider it again. Thank you very much for your helpful comment.

Modified Text: In the page 5, Line 219; Section No. 5

These Tregs are derived from naïve T cells through the action of tDCs, indicating that tDCs are involved in regulating immune responses.

  1. Correct to 'Tregs' (plural) on line 219.

Reply

Thank you very much for your comments. As you recommend, we changed the term. Thank you very much for your valuable comment.

Modified Text: In the page 5, Line 223; Section No. 5

Although Tregs control systemic immune responses, we found that pulsing DCs with specific antigen increases the number of antigen-specific Tregs, suggesting that regulating the systemic immune responses of patients may increase the risk of pathogen infection [13-15].

  1. The opposition 'abnormal' - ' natural' on lines 248-249 is incorrect. Please select the appropriate terms.

Reply

Thank you very much for your comments. We changed the term to make more accurate sentence. Pease see below the “Modified text” and reconsider it again. Thank you very much for your helpful comment.

Modified Text: In the page 6, Line 253; Section No. 7

Historically, autoimmune disease has been regarded as an abnormal response. However, many studies have reported that autoimmune disease is a normal phenomenon, with self-reactive antibodies and cells present in all normal individuals [62].

  1. Change 'current' to 'currently' (line 298).

Reply

Thank you very much for your careful comments. As you recommended, we modified some sentences. Please see below the “Modified Text”. Thank you very much for your helpful comment.

Modified Text: In the page 6, Line 302; Section No. 7

Currently on-going clinical trials of tDCs are focusing on standardization of methods used to generate tDCs ex vivo.

  1. The last sentence 'Although tDC-based clinical trials are much smaller than those of immunogenic DCs, tDC-based treatments for autoimmune diseases may be a novel therapeutic strategy for the future' (lines 301-303) is illogical. Size of the trial does not correspond to novelty in any way. This sentence is also repeated in the end of Conclusions section (lines 326-328). Please correct and eliminate redundancy.

Reply

Thank you very much for your comments. As you recommended, we eliminated redundancy and modified the sentence. Please see below the “Eliminated text and Modified text” and reconsider it again. Thank you very much for your valuable comment.

Eliminated Text: In the page 7, Line 304-606; Section No. 7

Modified Text: In the page 8, Line 330-333; Conclusions section

tDC-based immunotherapy for autoimmune and inflammatory diseases has potential as a novel therapeutic strategy for the future.

  1. What kind of contribution is 'collected the presented idea' (line 336)? Maybe, analyzed data? There is no data collection here since this is a review. It's also unclear (see the major criticism at the start of this review) what kind of 'support from S.-Y.C., N.-C.J., J.-Y.S., and H.G.S.' was received over the writing of this relatively short manuscript. If the support was editorial, these individuals should be excluded from authors list and should be thanked in the Acknowledgements.

Reply

Thank you very much for your comment. As we wrote in the “Author contributions section” in our manuscript, all authors actually contributed to this study. J.-H.N., J.-H.L., and D.-S.L. selected the theme of the manuscript and wrote the manuscript. S.-Y.C., N.-C.J., J.-Y.S., and H.G.S. discussed the idea and helped to draft the manuscript. We would be very grateful if you reconsider the author contribution of this study.